# Carcinoma-Associated Fibroblasts Accelerate Growth and Invasiveness of Breast Cancer Cells in 3D Long-Term Breast Cancer Models

**DOI:** 10.3390/cancers16223840

**Published:** 2024-11-15

**Authors:** Kingsley O. Osuala, Joshua Heyza, Zhiguo Zhao, Yong Xu, Kamiar Moin, Kyungmin Ji, Raymond R. Mattingly

**Affiliations:** 1Department of Pharmacology, Wayne State University, Detroit, MI 48201, USA; kosuala@med.wayne.edu (K.O.O.); heyzajo1@msu.edu (J.H.); kmoin@wayne.edu (K.M.); 2Department of Electrical and Computer Engineering, Wayne State University, Detroit, MI 48201, USA; zhao.zhiguo@wayne.edu (Z.Z.); yongxu@wayne.edu (Y.X.); 3Department of Neurology, Henry Ford Health, Detroit, MI 48202, USA; 4Department of Pharmacology and Toxicology, Brody Medical School, East Carolina University, Greenville, NC 27834, USA

**Keywords:** triple-negative breast cancer (TNBC), carcinoma-associated fibroblasts (CAFs), 3D parallel cocultures, microfluidic devices

## Abstract

Interactions between breast cancer (BCa) cells and carcinoma-associated fibroblasts (CAFs) in the tumor microenvironment (TME) are critical for cancer development and progression and can mitigate responses to therapies. Historically, it has been difficult to replicate in vivo long-term interactions between BCa cells and CAFs for in vitro studies and to define the precise roles of CAFs in BCa cell progression. Our novel microfluidic-capable culture devices called TAME (tissue architecture and microenvironment engineering) devices enable us to study cell-cell interactions of human breast cancer (BCa) cells and human CAFs through their secretome in 3D cultures for extended periods, up to and beyond 70 days. Using these systems, we observed that CAFs enhance BCa cell progression to an invasive phenotype. Moreover, secretome-mediated reciprocal interactions of BCa cells and CAFs promote their migration toward each other, suggesting that targeting signaling pathways to mediate interactions between BCa cells and CAFs could be a potential therapeutic approach for the prevention of BCa progression.

## 1. Introduction

Fibroblasts are a crucial part of the tumor microenvironment (TME), playing roles in maintaining the homeostasis of nearby cells and matrix turnover [1]. In pathological conditions such as cancer, fibroblasts transform into a proinflammatory subtype known as cancer-associated fibroblasts (CAFs) [1]. These CAFs, derived from a fibroblast lineage, express pro- and anti-inflammatory cytokines, proteases, and growth factors that promote tumor progression (see reviews [1,2,3]). Along with endothelial cells, immune cells, adipocytes, and patho-chemical conditions like hypoxia and acidosis, CAFs contribute to cancer progression and tumor cell response to drug therapies (see reviews [1,2,3,4]). To study CAFs and develop therapies to counteract their effects on tumor progression, it is essential to understand how they are recruited to and interact with cancer cells. However, in vivo studies of spatiotemporal interactions between cancer cells and fibroblasts are scientifically challenging and cost-prohibitive [5].

Using our microfluidic-capable cell culture devices coined TAME (tissue architecture and microenvironment engineering) (Patent: US 10,227,556 B2; [6,7]), we have developed 3D BCa cell:fibroblast parallel cocultures to emulate the long-term paracrine interactions between BCa cells and CAFs [6], in which the secretome of each cell type are shared in a common media without physical interaction between the two cell types. Our 3D BCa–fibroblast parallel cocultures enable the examination of paracrine interactions over extended periods and allow for the evaluation of changes in secretome, cell migration, morphology, and other factors implicated in tumor growth (e.g., hypoxia and acidosis) in real time. In this communication, we will limit our focus to triple-negative breast cancer (TNBC) as effective targeted treatments are not currently available [8,9,10]. Here, we demonstrate that CAFs promote the invasiveness of TNBC cells via secretome-induced interaction. Our 3D parallel coculture models, designed to identify paracrine signaling involved in CAF-mediated breast cancer invasiveness, will serve as a valuable preclinical tool for discovering new therapeutic approaches.

## 2. Materials and Methods

Reagents: Reconstituted basement membrane (rBM; Cultrex^TM^ reduced growth factor basement membrane extract) was purchased from Bio-Techne (Minneapolis, MN, USA). All culture media were phenol red-free. Dulbecco’s Modified Eagle Medium (DMEM) and mammary epithelial growth medium (MEGM) were purchased from Lonza (Basel, Switzerland). Hyclone fetal bovine serum (FBS) was from Cytiva (Marlborough, MA, USA). DMEM/F12, horse serum, L-glutamine, acid-washed glass beads, and all other chemicals, unless otherwise stated, were purchased from MilliporeSigma (Burlington, MA, USA). CellTrace^TM^ Far red was purchased from ThermoFisher Scientific (Waltham, MA, USA). Lentiviral mCherry, RNeasy RNA extraction, and OneStep RT-PCR kits were purchased from Qiagen (Hilden, Germany). PCR primers were purchased from BioRad (Hercules, CA, USA). hTERT lentivirus transfection kit (LVP1130-Puro) was purchased from GenTarget Inc. (San Diego, CA, USA). VWR^R^ DNA molecular weight markers [100 base pair (bp) ladder] were purchased from Avantor (Radnor, PA, USA).

Cells and cell maintenance: We used human TNBC cell lines: MDA-MB-231, MCF10.DCIS, and HCC70. MDA-MB-231 and HCC70 were purchased from ATCC and MCF10.DCIS [11] was obtained from the Biobanking and Correlative Sciences Core of the Karmanos Cancer Institute (Detroit, MI, USA). MDA-MB-231 and HCC70 were maintained in DMEM supplemented with 10% FBS. MCF10.DCIS was maintained in DMEM/F12 supplemented with 5% horse serum. We used human-derived fibroblasts provided by Dr. Simon W. Hayward (Northshore University HealthSystem Research Institute, IL, USA): CAF40T fibroblasts were derived from biopsy tissue diagnosed as invasive carcinoma and NAF98 fibroblasts were derived from benign tissue. Fibroblasts from tissue samples were isolated using a previously published protocol [12,13]. These fibroblasts were immortalized in Dr. Bonnie Sloane’s lab (Wayne State University, Detroit, MI, USA) using the hTERT lentivirus transfection kit according to the manufacturer’s instructions and were designated CAF40TKi and NAF98i, as in our previous reports [14,15]. Fibroblast cell lines were further characterized for their gene expression of Interleukin-1 beta, transforming growth factor beta, and CXCL12 (Appendix A; see [16,17,18]). All patient-derived cells were received de-identified and are therefore exempt from IRB oversight. Cell maintenance for fibroblasts is detailed in our previous reports [14,15]. All cell lines were authenticated through the genotyping service of the Karmanos Biobanking and Correlative Sciences Core (Detroit, MI, USA), and were verified to be Mycoplasma-free by periodic PCR testing.

Three-dimensional (3D) culture: We used a rBM overlay model optimized by Dr. Bonnie Sloane’s lab for live-cell imaging by confocal microscopy [6,7,15]. A seeding ratio of 5 TNBC cells and 1 CAF was used for 3D cultures as previously reported [14,15]. Details of our 3D long-term cultures were described previously [6]. Briefly, MDA-MB-231 TNBC cells (4 × 10^3^ cells) or CAFs (0.8 × 10^3^ cells) suspended in 15 μL of culture media were seeded on the top of the solidified 100% rBM (15 mg/mL rBM) in the center of each well, overlaid with 1:1 mixture of MEGM and fibroblast media in 2% rBM, and grown in TAME devices with separate (for monocultures) or linked wells (for parallel cocultures). In total, 100 μl of fresh 2% rBM overlay per well was added to cell cultures every 7 days throughout the culture period. MDA-MB-231 cells were transduced with mCherry to distinguish the TNBC cells from fibroblasts (labeled with CellTrace^TM^ Far red) in 3D cocultures according to our published procedures [6,7]. HCC70 and MCF10.DCIS cell lines were unlabeled and overlaid in 2% rBM with DMEM/F12 media containing 2.5% FBS.

Microbeads saturated with CAF-conditioned media (CM): CAF-CM were obtained from 24 h conditioning with CAFs at ~70% confluence on a 100 mm culture plate as described previously [15]. Approximately 100 acid washed glass microbeads in 10 microliters of PBS were incubated in 200 microliters of CAF-CM overnight at 4 °C on ice to allow the surface saturation of CAF-CM soluble factors to the microbeads. The beads in CAF-CM were acclimated for 30 min at 37 °C, and then added to the day-2 3D cell culture of DCIS cells. The 3D culture in the TAME device was then placed in a confocal microscope with a stage-top incubator at 37 °C and 5% CO_2_, and imaged live for a period of 24 h.

TAME devices: Details of the fabrication of TAME devices were provided in our previous report [7].

Image acquisition for quantitative analysis in 3D: Optical sections of a single field or 16 contiguous fields through the entire depth of the 3D structures were acquired every 7 days using either a Zeiss LSM 410 or a 780 confocal microscope (Carl Zeiss Microscopy, Jena, Germany). Images were reconstructed in 3D using Volocity software 7.0.0 (PerkinElmer, Waltham, MA, USA) as previously described [7,19,20]. X (green), y (red), and z (blue) arrows in the bottom left of images indicate the orientation of 3D images.

Area measurement of spheroid cores and invasive outgrowths: The area of invasive outgrowths from BCa spheroids was measured using ImageJ 1.54k as previously reported [21,22], and was calculated as follows: area of invasive outgrowths = total area of a spheroid (blue outer line)—area of spheroid core (yellow outer line) (Appendix A).

RT-PCR: Total RNA from 3D fibroblast cultures was isolated using RNeasy RNA extraction kits according to the manufacturer’s protocol. A reverse transcription reaction and PCR were performed using the OneStep RT-PCR kit following the manufacturer’s directions.

## 3. Results

### 3.1. Long-Term Paracrine Interactions of MDA-MB-231 Cells with Fibroblasts

To investigate secretome-mediated reciprocal interactions between TNBC cells and fibroblasts over an extended culture period, we performed 3D parallel cocultures of MDA-MB-231 (hereafter called 231) cells, a highly invasive TNBC cell line, and fibroblasts from normal breast tissues (NAF98i; hereafter called NFs) or invasive carcinoma (CAF40TKi; hereafter called CAFs) over a 70-day culture period in the TAME devices with linked wells, as illustrated in Figure 1A [6]. We compared the growth and invasive phenotypes of 231 cells grown in monoculture to those grown with NFs (Figure 1B) or CAFs (Figure 1C) in parallel cocultures. Within 7 days of seeding, 231 spheroids formed, and cell proliferation was observed in both monocultures and parallel cocultures. By day 28, the spheroidal tumor structures in the 231:CAF parallel cocultures were markedly larger than those of the 231:NF parallel cocultures and the monocultures (Appendix A). At days 47–56, invasive outgrowths from 231 spheroids in the 231:CAF parallel cocultures are apparent and begin to fill the matrix between spheroids, as seen at week 10 (Figure 1C and Appendix A).

We next tracked the distribution and migration of each cell type within their respective wells, in monoculture and parallel cocultures. Schematics in Figure 2A illustrate the distribution, size, and acquisition of the invasive phenotype of 231 spheroids at indicated times in 3D parallel cocultures of 231 cells and NFs or CAFs. In 231 monocultures, spheroids formed within 7 days and did not show marked alteration in size, morphology, or migration from their seeding location over the culture period, as illustrated in Appendix A. In parallel cocultures, we observed a significant migration of CAFs, but not NFs, toward 231 spheroids in neighboring wells (Figure 2A,B). A zoomed imaged of 231:CAF intermixed at day 63 shows the size and vast number of invasive outgrowths from the 231 spheroids in greater detail (Figure 2C). These results suggest that paracrine interactions between 231 cells and CAFs results in the increase in invasive phenotypes with spheroid size and the recruitment of CAFs to the cancer cell microenvironment. This is consistent with a previous report that 231 cells are a TNBC subtype that is characterized by stromal infiltration [23].

### 3.2. Long-Term Paracrine Interactions of DCIS or HCC70 Cells with CAFs

Having migrated to 231 spheroids by days 56–70, we acknowledge that the direct contact between cancer cells and CAFs likely enhances the migration and growth of the cancer cells, as previously described [14,15]. We next wanted to determine whether this was specific to the 231 cells or whether the results were translatable to other TNBC cell lines. Here, we observed similar phenotypic changes in TNBC:CAF interactions over a 50-day culture period using the pre-invasive MCF10.DCIS TNBC cell line [hereafter called DCIS; Figure 3(A1,B1)] as well as with the TNBC cell line, HCC70 [Figure 3(A2,B2)] in 3D parallel cocultures. DCIS and HCC70 spheroids in monocultures formed spheroidal structures (Figure 3A) that are consistent with others’ and our previous findings [12,18,24,25]. However, they exhibit features of a highly invasive phenotype with increased growth in size and invasive outgrowths when in parallel cocultures with CAFs (Figure 3B) as compared to monocultures at day 50 of the culture period. The observed increase in TNBC spheroids in 3D parallel cocultures with CAFs suggests that CAF-derived secretome might affect TNBC migration or dissemination to other regions.

Next, we collected conditioned media from CAFs (CAF-CM) and generated microbeads saturated with CAF-CM to evaluate TNBC cell response. Here, soluble factors in the CAF secretome were saturated to the surface of microbeads, thus acting as a chemoattractant for the cancer cells. Interestingly, DCIS spheroids migrated towards the microbead (Figure 4, movie), demonstrating the CAF-CM-mediated recruitment of DCIS cells. In DCIS or HCC parallel coculture with CAFs, we found that the migrating front of BCa cells showed fewer invasive outgrowths (Figure 5B–F) and smaller spheroids (Figure 5H–L), compared to spheroids located in the center of the colony (Figure 5A,B). The migration of DCIS and HCC70 spheroids was slower than that observed in the 231:CAF parallel cocultures and we did not observe the intermixing of CAFs with DCIS or HCC70 cells before the termination of the cultures on day 60.

## 4. Discussion

The present study demonstrates the in vitro secretome-mediated reciprocal interaction of TNBC cells and CAFs over an extended culture period. By incorporating CAFs into the TME of our 3D BCa model, we enhanced our model to elucidate paracrine networks between these two cell types which will facilitate the discovery of novel druggable signaling pathways. We observed that CAFs promote the increase in size as well as invasive phenotype of 231 spheroids via their secretome over a 70-day culture period (Figure 1 and Figure 2). Similar spatiotemporal changes in TNBC:CAF interactions were recorded over a 50-day culture period with two other TNBC cell lines, pre-invasive MCF10.DCIS and invasive HCC70 cells (Figure 3). Additionally, we observed slight variations in the morphology of BCa spheroids near the migrating front of each colony as compared to the core of the colony (Figure 5). The migrating front comprised smaller spheroids with fewer outgrowths from each spheroid. We believe this to be a result of the rolling method of migration whereby the spheroids tumble directionally towards an attractant, in this case CAFs or CAF-CM-saturated microbeads, as shown in Figure 4B. Spheroids closest to the attractant implemented a rounder shape to facilitate faster migration through the matrix to reach the attractant. Taken together, these data indicate that CAFs have a role in the promotion of growth and invasiveness of TNBC spheroids via their secretome.

BCa progression is regulated by multiple processes including juxtracrine and paracrine interactions with neighboring fibroblasts. Understanding these interactions, comprising the induction of cytokine secretion [26], exosome release [27,28], and ECM remodeling [29], will aid in unlocking the complexed nature of cancer development. The present study provides insights into the initial recruitment and migration of CAFs and BCa cells through their paracrine interactions by tracking their spatiotemporal interactions in vitro for extended periods using our TAME devices. Our group has previously identified and validated paracrine roles for hepatocyte growth factor (HGF) and interleukin-6 (IL-6) secreted by CAFs in the growth and migration/invasion of TNBC cells, including pre-malignant DCIS [14,19]. CAFs present a promising therapeutic strategy based on these paracrine interactions with BCa cells; however, to date, most clinical trials have failed to improve clinical outcomes. This may be due to the functional heterogeneity of CAFs present in the TME [26,30,31,32,33]. We aim to utilize our TAME devices and 3D culture models to increase our understanding of their precise origins/characteristics and how they are initially activated in the BCa microenvironment.

Developing therapeutics through the phenotypic screening of tumor cells requires robust and quantifiable methods to ensure a high level of translational and practical impact. Our 3D cultures grown in the TAME devices enabled us to track changes in BCa cell morphology and distribution over longer periods of time versus conventional 3D cell cultures. The TAME device also allowed for the evaluation of interactions of BCa cells with fibroblasts either in monoculture or parallel coculture with CAFs or NFs. The current study and future studies aim to investigate adaptations in the secretome of TNBC cells in monocultures, serving as controls, or in parallel cocultures with either NFs or CAFs. This approach will provide insights to determine how CAFs affect the growth, invasive characteristics, and migration of TNBC spheroids. We believe that our 3D parallel coculture models may be a reliable pre-clinical model to better mimic in vivo cancer cell–TME interactions over extended periods.

## 5. Conclusions

The present study demonstrates that paracrine interactions of TNBC and CAFs play a critical role in promoting the invasive phenotype of TNBC spheroids. We also demonstrated the applicability of our novel TAME devices to studying the paracrine interactions between two cell types grown in 3D parallel cultures over time. Our results provide insights into how CAFs contribute to the increased size and invasiveness of TNBC spheroids and how paracrine interactions between TNBC cells and CAFs encourage migration towards each other over an extended culture period. Our findings further support targeting paracrine pathways between TNBC cells and fibroblasts as a viable methodology to slow or prevent the progression of TNBC.

## 6. Patents

The present study is associated with the U.S. Patent US10227556B2.

## Figures and Tables

**Figure 1 cancers-16-03840-f001:**
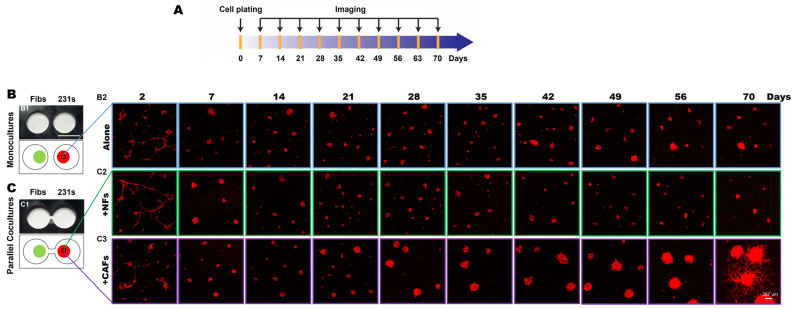
CAFs promote an invasive phenotype of 231 spheroids in 3D parallel cocultures. (**A**) Schematic illustration of the experimental timeline. (**B**,**C**) En face views of 3D reconstructions of MDA-MB-231 [231; mCherry-transduced (red)] monocultures (**B**) and parallel cocultures with fibroblasts (Fibs) (green), either normal fibroblasts (NFs) or carcinoma-associated fibroblasts (CAFs; CellTrace^TM^ Far red-labeled) at indicated times in the TAME device (**C**). Scale bar, 257 μm.

**Figure 2 cancers-16-03840-f002:**
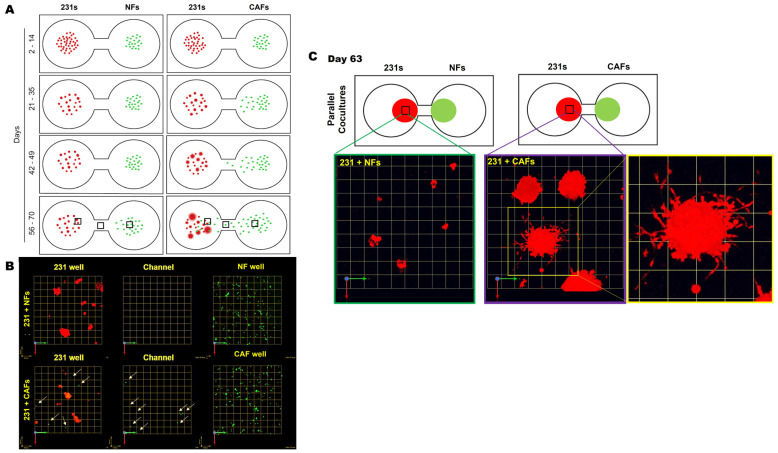
CAFs, not NFs, preferentially migrate toward 231 spheroids with an increase in invasive outgrowth and spheroid size in 3D parallel cocultures. (**A**) Schematics demonstrating observed spatial distribution, size of cell clusters, and the formation of invasive outgrowths from 231 cells at select time frames during the culture period. (**B**) Images from 63-day parallel cocultures of 231s with NFs (top row) or CAFs (bottom) at three locations across the TAME device. Note: 231 wells (left column), channel between wells (middle column), and fibroblast well (right column) [fibroblasts shown by green dots and tumor cells red]. Arrows highlight CAFs within 231 wells and in the channel linking wells. Each grid represents 51 μm. (**C**) Enlarged images of 231 spheroids in parallel cocultures with NFs (left) or CAFs (middle and right) at day 63 in greater detail; each grid represents 257 μm.

**Figure 3 cancers-16-03840-f003:**
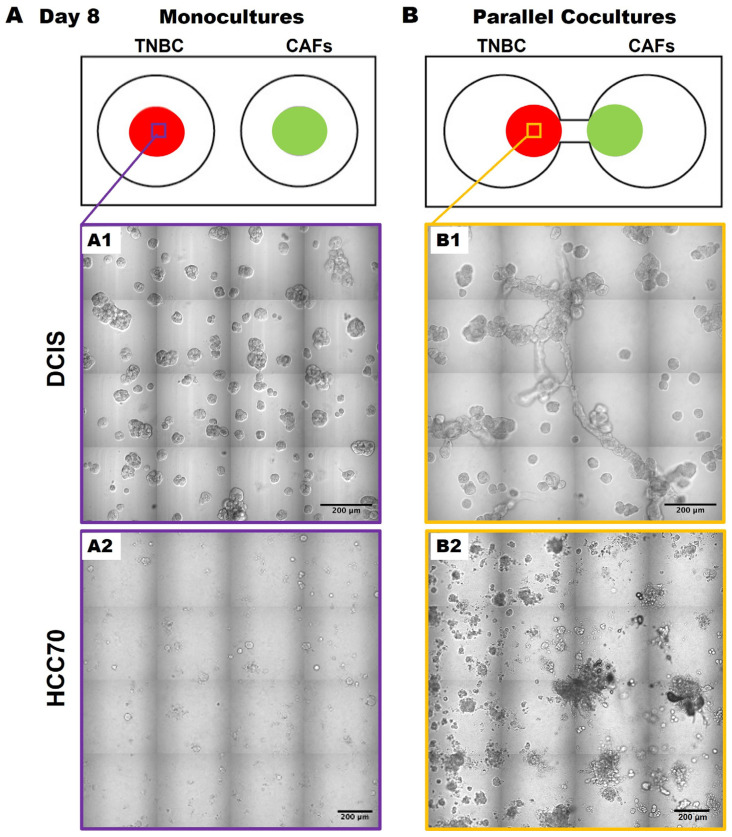
CAFs promote an invasive phenotype of MCF10.DCIS and HCC70 cells in 3D parallel cocultures. Schematics (top) and differential interference contrast (DIC) images of DCIS and HCC 70 cells in monocultures (**A**) and parallel cocultures (**B**) at day 8. Representative tiled DIC images of DCIS cells (**A1**) and HCC70 cells (**A2**) in monocultures. Tiled DIC images of DCIS cells (**B1**) and HCC70 cells (**B2**) in parallel cocultures with CAFs. Scale bars, 200 μm.

**Figure 4 cancers-16-03840-f004:**
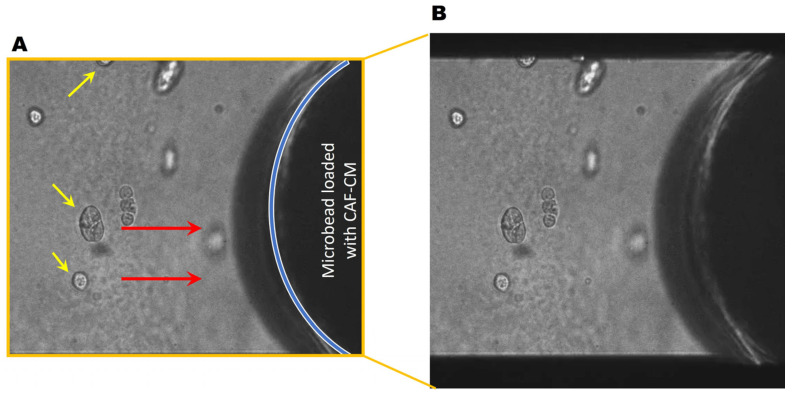
DCIS spheroids migrate towards CAF-CM. (**A**) Representative DIC images of the monoculture of DCIS spheroids (yellow arrows) near a CAF-CM-saturated microbead. Red arrows indicate the direction of migration. The location of the microbead is indicated by the blue line. (**B**) Video showing DCIS cells migrating towards a CAF-CM coated microbead over a period of ~20 h. Video available: https://zenodo.org/uploads/13924370 and also in the Appendix A.

**Figure 5 cancers-16-03840-f005:**
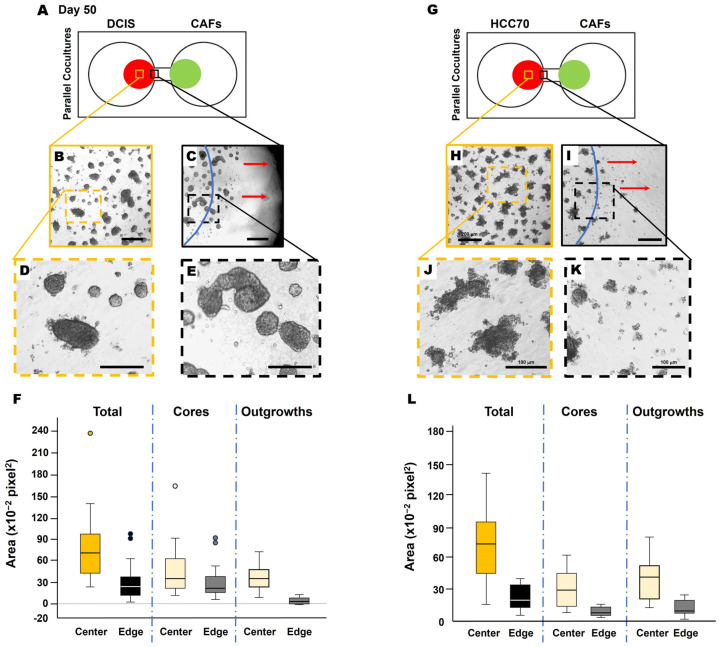
Migrating TNBC spheroids exhibit morphological variation from those closer to the collective center. Schematics of DCIS cells (**A**) and HCC70 cells (**G**) in parallel coculture with CAFs in the TAME device at day 50. Low magnification of DCIS structures (**B**,**C**) or HCC70 structures (**H**,**I**) in the collective center of TNBC colonies (**B**,**H**) and nearest to the interconnecting channel (**C**,**I**). Dashed orange squares in (**B**,**H**), and dashed black squares in (**C**,**I**) are enlarged in (**D**,**J**,**E**,**K**), respectively. The location of the initial cell plating is indicated by blue lines in (**C**,**I**). Red arrows in (**C**,**I**) indicate the direction of migration. Scale bars: (**B**,**C**,**H**,**I**), 200 μm; (**D**,**E**,**J**,**K**), 100 μm. Note that DCIS and HCC70 colonies in (**C**,**I**) at the nearest to the interconnecting channel appear to have smaller and more single cells free from spheroids. Spheroid area quantification of DCIS (**F**) and HCC70 (**L**), total spheroids, core of spheroids, and invasive outgrowths of spheroids. Measurements taken from the center of the colony (center; yellow boxes of (**A**,**G**) and at the migrating edge of the colony [edge; black boxes of (**A**,**G**)] were quantified with ImageJ 1.54k and presented in box-and-whisker plots.

## Data Availability

Data are contained within the article.

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
