# Peer review of "Carcinoma-Associated Fibroblasts Accelerate Growth and Invasiveness of Breast Cancer Cells in 3D Long-Term Breast Cancer Models"

_cancers, 2024, doi:10.3390/cancers16223840_

Round 1
Reviewer 1 Report
Comments and Suggestions for Authors
The manuscript is interesting but the section Materials and Methods should be improved:
1. It is necessary to describe the methods of the isolation of normal fibroblasts and CAFs.
2. Cells from one donor only were used. Because of heterogeneity between patients, it is necessary to use cells from more donors.
3. Cells were immortalized, but the procedure is not described. It is known that immortalization changes the cell behaviour. The use of primocultures should be better.
4. CAFs are heterogeneous (mCAF, iCAF , apCAF). The used cells must be characterised
Author Response
Reviewer #1
The manuscript is interesting, but the section Materials and Methods should be improved:
1. It is necessary to describe the methods of the isolation of normal fibroblasts and CAFs.
As requested, we now included more detail on isolation and immortalization of fibroblasts in Materials and Methods, lines 94-101.
2. Cells from one donor only were used. Because of heterogeneity between patients, it is necessary to use cells from more donors.
We agree with the reviewer’s comment. In our previous works, we have observed that CAFs from 4 different sources showed a minimal difference in protease expressions (Koblinski JE et al., J Biol Chem, 2002, PMID: 12072442) and increased in the growth and invasiveness of breast tumor cells (Rothberg JM et al., Biochim Biophys Acta, 2013, PMID: 21854877). In a future study, we aim to examine the spatio-temporal interactions of TNBC:CAF with different human CAF subtypes (S1-S4; Costa A et al., Cancer Cell, 2018, PMID: 29455927).
3. Cells were immortalized, but the procedure is not described. It is known that immortalization changes the cell behaviour. The use of primocultures should be better.
We have included detail on the isolation and immortalization of fibroblasts in Materials and Methods, lines 94-101. We understand for the reviewer’s concerns on immortalization. Based on our experience, the use of primoculture was not practical and provided limited utility under these conditions. Given the 3D in vitro cell culture methodology and the necessity to fluorescently label cells, there are several
factors that likely alter cell population behavior from that of the in vivo state. We understand and state that these studies are only a surrogate for in vivo conditions and that they have limitations as do all studies.
4. CAFs are heterogeneous (mCAF, iCAF, apCAF). The used cells must be characterized.
We agree with the reviewer’s comment. Based on gene expression as shown in our previous report (Osuala KO et al., BMC Cancer, 2015, PMID: 26268945), the CAFs we used for the present study fall into the iCAF or CAF-S1 subtype (Costa A et al., Cancer Cell, 2018, PMID: 29455927). In a future CAF subtype-specific study, we will include a proteomics array to characterize several CAF cell lines.

Reviewer 2 Report
Comments and Suggestions for Authors
The authors describe a novel method of investigating the paracrine effects of CAFs on TNBC breast cancer cell migration and for the most part, their evidence is compelling, however there are a few things that could be improved.
1. They talk about "secretome-mediated" effects, but it could be argued that beyond that, the effects on invasion could be cell-cell interaction-mediated (i.e., the secretome induces migration towards but then once there, it could very well be cell-cell interactions between the two cell types that mediates BCa invasion).
2. They don't provide any statistical analysis of the extent of increased invasion - a simple ImageJ multipoint tool analysis could help here.
3. They talk about increase in growth, but I'm not convinced it's not just larger spheroids being formed. There is no analysis of proliferation or viability or even simple ImageJ analysis of spheroid size to corroborate this statement.
4. Are they confident that after 70 days in culture in these TAME devices the cells are still alive and well, behaving normally? Do they refeed them? How and how often? This needs to be included in the methods.
5. In the methods (and/or results section) it would be helpful if they could stipulate what happens when they soak the microbeads in conditioned medium - do they absorb it and therefore act as a reservoir for secreted factors? Or do the factors in the secretome adhere to the surface of the beads acting as ligands to cell receptors?
6. Their claims for Figure 5 (particularly 5F) are perhaps overstated. I'm not convinced that the spheroids nearer the microbeads show "fewer invasive outgrowths" - again, some ImageJ analysis to quantify this would help.
7. The discussion is far too brief and mostly self-cites. This needs to be expanded to include thorough discussion of other studies in the field. I would also expect to see a more thorough discussion of the potential factors in the CAF secretome that could be responsible for these effects.
8. On that note, it would be ideal if they could do some analysis of the contents of the secretome - has this been done by them or anyone else before which could lend credence to their discussion?
9. It's quite light on references, but this will be improved on addressing point 7 above.
10. There are a few minor typos (see attached copy of the manuscript and look for the red writing). Also, note for the editors, the DOI for the videos isn't working - it takes me to an "Error: DOI Not Found" page.

Author Response
Reviewer #2
The authors describe a novel method of investigating the paracrine effects of CAFs on TNBC breast cancer cell migration and for the most part, their evidence is compelling, however there are a few things that could be improved.
1. They talk about "secretome-mediated" effects, but it could be argued that beyond that, the effects on invasion could be cell-cell interaction-mediated (i.e., the secretome induces migration towards but then once there, it could very well be cell-cell interactions between the two cell types that mediates BCa invasion).
We agree with the reviewer’s comment. We do not exclude the possibility that direct contact between tumor cells and fibroblasts stimulate an increase in their migration and invasiveness once they are in physical contact. We have included this description in the Result, lines 213-215.
2. They don't provide any statistical analysis of the extent of increased invasion - a simple ImageJ multipoint tool analysis could help here.
Thank the reviewer for the suggestion. We have measured the area of core and invasive outgrowth from spheroids using ImageJ and have included the data in Supplemental Figures 2, 3 and Figure 5F, L.
3. They talk about increase in growth, but I'm not convinced it's not just larger spheroids being formed. There is no analysis of proliferation or viability or even simple ImageJ analysis of spheroid size to corroborate this statement.
TNBC spheroids formed by TNBC cells in our 3D cultures increased the invasive phenotype and sizes of the spheroids in parallel cocultures with CAFs. In this manuscript, we assessed changes in volume, invasive phenotype, and distribution of TNBC spheroids by live-cell imaging techniques. These techniques do not require termination of the cultures. The spheroid volumes are an alternative measure
of proliferation as we determined by quantification of spheroid volumes including cores and invasive outgrowth. Counting nuclei, Ki-67 staining, or Live/Dead assay is an endpoint assay conducted on cultures that are not live and thus is not compatible with the spatiotemporal analyses we performed. As the reviewer suggested, we have added the measurement of area of core and invasive outgrowth from spheroids using ImageJ in Supplemental Figures 2, 3 and Figure 5F, L.
4. Are they confident that after 70 days in culture in these TAME devices the cells are still alive and well, behaving normally? Do they refeed them? How and how often? This needs to be included in the methods.
Since we observed the continuous changes in distribution and size of TNBC spheroids and CAFs over the culture period, we are confident that the cells are alive. We have described the details in our previous report (see a reference #6 on lines 366-367) and have added the details in Materials and Methods, lines 118-124. Our future studies will evaluate viability and proliferation of TNBC spheroids as well as secretome analysis.
5. In the methods (and/or results section) it would be helpful if they could stipulate what happens when they soak the microbeads in conditioned medium - do they absorb it and therefore act as a reservoir for secreted factors? Or do the factors in the secretome adhere to the surface of the beads acting as ligands to cell receptors?
As requested, we have added the details to clarify that the microbead surface is saturated with soluble factors from the CAF-CM in Materials and Methods, lines 130-133.
6. Their claims for Figure 5 (particularly 5F) are perhaps overstated. I'm not convinced that the spheroids nearer the microbeads show "fewer invasive outgrowths" - again, some ImageJ analysis to quantify this would help.
We agree with the reviewer’s comment. We have corrected the statements in the Result, lines 261- 264, and have included the area measurement of core and outgrowth of spheroids in Figure 5F, L.
7. The discussion is far too brief and mostly self-cites. This needs to be expanded to include thorough discussion of other studies in the field. I would also expect to see a more thorough discussion of the potential factors in the CAF secretome that could be responsible for these effects.
We agree that we could have done a better job on references and discussion. We have cited more references relevant the present study and have added more discussion on CAF secretome and BCa progression under the Discussion, lines 277-284 and 286-300.
8. On that note, it would be ideal if they could do some analysis of the contents of the secretome - has this been done by them or anyone else before which could lend credence to their discussion?
We have collected 50 μl of conditioned media (CM) every 7 days for future secretome analysis. We have recently been in contact with scientist at O-Link regarding a proteomic analysis methodology with high sensitivity from a small volume of culture media. We are exploring the potential of this new technology to evaluate the secretomes of our 3D cultures. We aim to define the secretome that coincides with spatio-temporal changes in migration and invasive phenotype in our 3D culture system.
9. It's quite light on references, but this will be improved on addressing point 7 above.
We have cited more references relevant the present study.
10. There are a few minor typos (see attached copy of the manuscript and look for the red writing). Also, note for the editors, the DOI for the videos isn't working - it takes me to an "Error: DOI Not Found" page.
We have corrected typos and grammatical errors. We have asked Ms. Dana Dong, the assigned editor in Cancers, to confirm the proper upload procedure for the movie file.

Reviewer 3 Report
Comments and Suggestions for Authors
The authors have tried to be successful in establishing a novel microfluidic based in vitro system to assess the cancer properties in a 3-D manner keeping in mind the interactions between cancer cells and the cells of the TME. This would serve as an excellent tool for the researchers to assess their cancer related studies prior to an in vivo approach without compromising the cell-cell interactions in a living hist scenario. The research is written and presented well with adequate information about materials and methodology.
The paper can be accepted in the present form.
Author Response
Reviewer #3
The authors have tried to be successful in establishing a novel microfluidic based in vitro system to
assess the cancer properties in a 3-D manner keeping in mind the interactions between cancer cells
and the cells of the TME. This would serve as an excellent tool for the researchers to assess their
cancer related studies prior to an in vivo approach without compromising the cell-cell interactions in a
living hist scenario. The research is written and presented well with adequate information about
materials and methodology.
The paper can be accepted in the present form.
We thank the reviewer for the positive comments.

Round 2
Reviewer 1 Report
Comments and Suggestions for Authors
The manuscript was carefully revised and it is suitable for acceptance yet.